# Taxes to red and processed meat to promote sustainable and healthy diets in Mexico

Kaela Connors[1,2]*, Juan A. Rivera[3], Peter Alexander[1,2], Lindsay M. Jaacks[2], Carolina Batis[4], Dalia Stern[5], Martín Lajous[3,6], M. Arantxa Colchero[7]

1 School of Geosciences, University of Edinburgh, Edinburgh, United Kingdom, 2 Global Academy of Agriculture and Food Systems, University of Edinburgh, Midlothian, United Kingdom, 3 Center for Research in Population Health, National Institute of Public Health, Cuernavaca, Morelos, Mexico, 4 Center for Nutrition and Health Research, National Institute of Public Health, Cuernavaca, Morelos, Mexico, 5 SECIHTI-Center for Research on Population Health, National Institute of Public Health, Cuernavaca, Morelos, Mexico, 6 Department of Global Health and Population, Harvard T.H. Chan School of Public Health, Boston, Massachusetts, United States of America, 7 Center for Evaluation and Surveys, National Institute of Public Health, Cuernavaca, Morelos, Mexico

* k.m.connors@sms.ed.ac.uk

## Abstract

### Background

Consumption of red and processed meat is above sustainable and healthy dietary targets in Mexico. Policies to promote greater adherence such as taxing meat are needed to reduce consumption. Here, we evaluated how price increases to red and processed meat could shift consumption for meat as well as other key food groups.

### Methods

Using data from the Mexican National Household Income and Expenditure Survey (2018, 2020, 2022), we estimated own- and cross-price elasticities of demand for 10 food groups. These were used to predict changes in quantity demanded of each food group according to price increase scenarios.

### Results

Price increases to meat increased demand for substitutes such as fruits and vegetables, legumes, poultry, and seafood, and reduced demand for salty snacks, sweets and sugary beverages. Substantial increases to the price of processed meat markedly reduced processed meat consumption, increased consumption of protein- and nutrient-rich foods, and resulted in more modest levels of red meat consumption. Lower-income groups were more sensitive to price increases but still met dietary recommendations for meat and substituted these foods with alternatives such as legumes and seafood.

**Data availability statement:** The household income and expenditure data are publicly available through the Mexican National Institute of Statistics and Geography (INEGI) https://en.www.inegi.org.mx/programas/enigh/. The national diet data are publicly available through the National Health and Nutrition Survey (ENSANUT) https://ensanut.insp.mx/. Data for the longitudinal cohort of middle-aged Mexican women (Mexican Teacher's Cohort) is de-identified since it contains private information and is housed at the Mexican National Institute of Public Health (INSP). Ethical approval was granted by INSP. Completing the baseline questionnaire was considered consent (Project CI: 1645). Authors received official permission to access, analyze, and share findings using the data for the purposes of this study. Researchers interested in gaining access to the data may do so upon request to esmaestras@insp.mx.

**Funding:** Kaela Connors received financial support from the Edinburgh Earth, Ecology, and Environment Doctoral Training Partnership (E4 DTP). The funders had no role in study design, data collection and analysis, decision to publish, or preparation of the manuscript.

**Competing interests:** The authors have declared no competing interests exist.

## Conclusions

Increasing the price of red and processed meat through a tax may promote greater adherence to sustainable and healthy dietary targets in Mexico. It simultaneously increased demand for healthier substitutes, and reduced demand for unhealthy complements. Substantially raising the price of processed meat only may be an effective strategy for addressing diet-related risk factors, while delivering environmental benefits. Additionally, meat taxes have the potential to promote improvements in diet quality and equitable health outcomes.

## Introduction

Government-supported demand-side interventions, such as fiscal policies, are one way to support food system transformations and behavior change [1]. Food-specific taxes have been most commonly applied to sugar-sweetened beverages (SSBs), nonessential energy-dense foods, and saturated fat to address diet-related chronic disease burdens [2–4]. Taxes have indeed been effective in reducing targeted food purchases [4,5], therefore, may serve as a policy tool to achieve double-duty goals for health and sustainability. Estimating price elasticities enhances empirical understanding of demand responses to price increases of taxed commodities as well as potential substitutes and complements to those commodities, and thus have been calculated to inform tax policy design [6,7].

Meat production accounts for a majority of food system emissions, responsible for one third of greenhouse gas emissions [8]. Simultaneously, high consumption of red and processed meat is also associated with colorectal cancer [9] and type 2 diabetes [10], among other health risks [11,12]. Moderating levels of meat consumption is central to sustainable and healthy diets [13,14], particularly in high- and middle- income contexts where meat consumption is disproportionately high [15,16]. Several studies have estimated or utilized price elasticities to model hypothetical taxes on meat in Europe, the United States, and more recently, Brazil for precisely this purpose [17–21]. Demand responses to price vary by population and commodity; curbing demand for red and processed meat has been found to require more aggressive approaches such as higher rates of taxation as well as varying rates across consumer groups and geographies [17,18]. Thus, derivation of price elasticities to estimate demand responses to such policies is a practical tool that may inform the magnitude of increases that would be needed to achieve desired consumption levels.

Global diets require drastic transformations to keep within net zero emissions targets and alleviate significant diet-related disease burdens [8,22]. Recognizing this, Mexico integrated sustainability into the national Dietary Guidelines in 2023 [14]. Nevertheless, adherence to sustainable and healthy diets remains low in Mexico [23,24] despite research suggesting that such diets are affordable in this context [25,26]. In particular, meat consumption continues to increase and is well above recommended levels [23]. Thus, reducing red and processed meat consumption has been identified as key to increasing the sustainability of healthy diets in the Mexican

context [23,24]. Mexico was one of the first countries to pass a food-specific tax in 2014 on SSBs and an ad valorem tax on non-essential energy-dense foods motivated by increasing diet-related chronic disease burdens such as obesity and type 2 diabetes [5]. The path toward a sustainable and healthy diet in Mexico has been clearly outlined [14] and assessed for feasibility and affordability [26], but an implementation gap remains that necessitates government policy intervention. Building on Mexico's food policy toolkit [1], a tax to red and processed meat can help address this gap. To date, such a tax has not been studied in the Mexican context and understanding of how it might shift consumption in the rest of the diet – essential to diet quality – is not well understood.

We assessed how to achieve sustainable, equitable, and healthy dietary targets for red and processed meat consumption in Mexico. The four specific aims were to (1) estimate own- and cross-price elasticities of red and processed meat, (2) estimate demand responses to price increases in red and processed meat under various scenarios and apply them to consumption data, (3) explore if price elasticities varied by household income and other effect modifiers (i.e., survey year, and consumption status) and (4) corroborate cross-sectional findings by evaluating dietary substitutions and complements to red and processed meat in a longitudinal cohort of Mexican women.

## Materials and methods

### Price elasticities for red and processed meat

We used 2018, 2020, and 2022 data from the Mexican National Income and Expenditure Survey (ENIGH, by its Spanish acronym), administered by the Mexican National Institute of Statistics and Geography. The survey has a two-stage probabilistic design and collects cross-sectional, nationally representative information on household income and expenditures including sociodemographic information every two years. It consolidates households' daily food and beverage expenditures for one week, as reported by the head of household. After excluding households that did not report food expenditures or that only reported non-descript meal events outside the home, the unweighted analytical sample for all three years combined was 248,226 (S1 Text).

Food expenditures were categorized into 11 food groups: (1) dairy, (2) eggs, (3) salty snacks, sweets, and sugary beverages (herein, discretionary foods), (4) seafood, (5) fruits and vegetables (F&V) (6) legumes, nuts and seeds (herein, legumes), (7) grains, roots, and tubers (herein, grains), (8) unprocessed poultry and offal (herein, poultry), (9) processed meat (including processed red meat and poultry), (10) unprocessed red meat and offal (herein, red meat), and (11) other foods (i.e., spices, condiments, oils, non-sweetened beverages such as water, tea, and coffee) (S1 Table in S1 File). These food group categories were chosen to reflect food groups of the EAT-*Lancet* and 2023 Mexican Dietary Guidelines with a few adjustments given the high prevalence of consumption of certain food groups (i.e., eggs) [13,14] (S1 Table in S1 File).

### Statistical analyses

We calculated uncompensated own- and cross-price elasticities of demand for 10 food groups through estimation of a complete demand system using a linear approximate of an Almost Ideal Demand System (LA/AIDs) [27]. The linear approximate model was chosen to model a linear relationship between price and quantity demanded for interpretability and consistency with prior food literature [27–29]. Briefly, the model calculates a set of estimates based on the proportion of the budget dedicated to a particular food group out of the total household food expenditures. These estimates approximate the response in demand to changes in the price of a food group without considering changes in income.

Price was calculated based on the quantity purchased (kg or liters) and the expenditures reported for that quantity of the food group. Prices were then averaged at the municipality level to reduce potential record biases at the household level [6]. Outliers at the municipality level that fell ±2 standard deviations from the national mean were replaced with the weighted average national price for that food group (n = 276 [0.4%] for 2018; n = 166 [0.2%] for 2020; and n = 138 [0.2%] for 2022). The expenditure share of each food group was calculated as the sum of expenditures in each food group divided by total household food expenditure.

The LA/AIDs model is defined as:

$$\underbrace{w_{hgmt} = \alpha_g + \sum_{j=1}^{j} \beta_{gi} \ln p_{mjt} + \gamma \ln \frac{E}{P_*}}_{\text{LA/AIDs model}} + \underbrace{\sum_{v=1}^{v} \delta_{gv}\eta_{hmtv}}_{\text{Covariate term}} + \mu_{hmgt}$$

(1)

As specified, $w_{ghmt}$ is the proportion of household expenditures spent on a particular food group $g$ for household $h$ residing in municipality $m$ in survey year $t$. Each individual food item within each food group is denoted as $j$ and $p_{mjt}$ represents the price of each food item at the municipality level $m$ in the year $t$ the survey was administered. $E$ represents total household expenditures on food while lnP* is the Laspeyres price index represented as $\ln P_{jt}$ in Equation 1 above. The covariate term $\eta$ denotes shifter variables which may affect demand for a good, $v$, at the household and municipality level that will be added to the model. Lastly, $\mu$ is the error term for the model.

We included the Laspeyres Price Index to account for changes in price over time and maintain linearity in the parameters by weighting price according to baseline prices and shares (in this case, 2018) for a constant basket of foods [30]. The index is calculated as:

$$\ln P_{jt} = \sum_{g=1}^{j-1} \overline{w_g} * \ln p_{mjt}$$

$p_{mjt}$ is defined as the price per unit for each $jth$ food item in municipality $m$ at survey year $t$. In addition, $\overline{w_g}$ is defined as the average budget share for the food group $g$ pertaining to food item $j$. More detailed information on the price elasticity estimation can be found in S1 Text.

The model was estimated using Ordinary Least Squares (OLS) equation and 'other foods' was omitted to account for potential collinearity between food groups. Estimation using OLS assumes that errors are normally distributed and that there is no correlation in errors between equations [27]. We also restricted the model to comply with the homogeneity and symmetry constraints, and verified that the adding-up property was met [27]. We accounted for potential sociodemographic factors that could influence the relationship between price and demand through accounting for demand shifters in the model. These included educational attainment of the head of household, year of survey (2018, 2020, and 2022), region of residence (urban vs rural), and adult equivalent based on estimations derived from the Mexican population considering age and size composition of the household [31]. Adult equivalent has been previously included in food demand system analyses for Mexico [6] to account for how household composition may impact welfare due to household composition and the distribution of income amongst members. Price elasticities were first presented by pooling three survey rounds. We then calculated price elasticities stratified by: (1) survey round, (2) income quintile, (3) consumers of red meat (non-zero expenditures on red meat), and (4) consumers of processed meat (non-zero expenditures on processed meat) to estimate potentially heterogenous impacts on purchases associated with price increases on different consumer groups. More details on model specifications are in S1 Text. We used the package for *Demand Analysis with the Almost Ideal Demand System* in R statistical software to estimate the demand system [32].

We tested price increases previously modeled in other settings and chose combinations where processed meat prices were higher than red meat due to more consistent evidence of processed meat's adverse health effects [17,33] (S1 Text; S1 Fig in S1 File).

Price Increase Scenarios:

1) Reference: no price increase

2) Scenario 1: price increase of 30% to both red and processed meat [19,20]

3) Scenario 2: 15% price increase to red meat and 30% to processed meat [19]

4) Scenario 3: 7% price increase for red meat and 47% to processed meat [17]

5) Scenario 4: 100% price increase to processed meat only [17]

We then obtained percent changes in quantity demanded of food groups included in the demand system by applying the price increases to meat and then calculating the change in demand of food groups using own- and cross-price elasticities. We performed the same procedure within each subpopulation.

We conducted two sensitivity analyses. In the first, we separated meat into 'expensive' and 'cheaper' cut of meat according to price instead of red and processed meat given the heterogeneity of prices within the meat category and calculated price elasticities (S1 Text; S2 Table in S1 File). In the second, we estimated the demand system using the Seemingly Unrelated Regression [34] – which accounts for correlation in the error terms across food group equations – to compare with results estimated from OLS, that omitted the 'other foods' group, to ensure that our estimations of the price elasticities of demand were robust.

## Application to national consumption data

Baseline consumption data were from a 24-hour dietary recall administered in 2016 by the Mexican National Health and Nutrition Survey (ENSANUT, by its Spanish acronym). ENSANUT uses a multi-stage probability design to sample the civilian, non-institutionalized population of Mexico. Data are obtained through face-to-face interviews performed in participants' homes. The analytic sample for adults (≥ 18 years) who had completed the recall, were not pregnant or lactating, and had plausible energy values (total energy intake ≥400 kcal/day and ≤ 4,000 kcal/day) was 1,558 participants.

We categorized red and processed meat into their respective groups, including meat consumed in mixed dishes which were disaggregated using a recipe file developed by the National Institute of Public Health, Mexico (S3 Table in S1 File). We also created 'expensive' and 'cheaper' cuts of meat categories using dietary consumption data from ENSANUT by matching items included in each category from the expenditure data when possible (S4 Table in S1 File).

The estimated changes (%) in quantity demanded of red and processed meat were translated to reductions in daily intake (g) for the Mexican adult population and subgroups (i.e., red meat consumers, processed meat consumers, lowest income, highest income) (S1 Text). Consumption after price increases were applied was compared to dietary targets stipulated by the 2023 Mexican Dietary Guidelines (approximately 30g/day for red meat and 0g/day for processed meat) and the EAT-*Lancet* (approximately 14 g/day for red meat and 0 g/day for processed meat) based on an average adult consuming 2,500 kcal/day [13,14]. We also calculated consumption of animal-source protein foods (the sum of red meat, poultry, eggs, and fish) before and after price increases to compare them to the Mexican and EAT-Lancet targets (140g/day and 84g/day, respectively S1 Text).

## Dietary substitutions using longitudinal dietary intake data

To corroborate cross-sectional findings, we conducted a secondary analysis of a longitudinal cohort of Mexican women in the states of Jalisco and Veracruz. Data were from the Mexican Teachers' Cohort (MTC), a prospective cohort study of 115,307 middle-aged female teachers across Mexico. MTC evaluated dietary intake using a validated 140-item food frequency questionnaire (FFQ). The sample size was 1,417 based on those who had completed the FFQ in all three waves where dietary data was collected (2006, 2008, and 2014) and based on additional eligibility criteria (S1 Text). We categorized food groups according to those modeled in the demand system (S1 Text; S5 Table in S1 File). Information on egg consumption was not collected in 2006 and therefore was not included.

We calculated changes in quantity consumed (g/day) for each period (Period 1: 2006–2008 and Period 2: 2008–2014). We then evaluated potential dietary substitution and complement behavior by applying a linear regression predicting changes in red, processed, and total meat intake with changes in non-meat food groups adjusting for changes in total daily energy intake (S1 Text).

## Results

The own-price elasticities of red and processed meat in the pooled sample after adjustment were −0.79 and −0.83, respectively (Table 1; S2 Text; S6–S11 Tables in S1 File). In other words, a 10% increase in the price of red meat would result in a 7.9% decrease in quantity demanded of red meat, and a 10% increase in the price of processed meat would result in an 8.3% decrease in quantity demanded of processed meat (Table 1). Own-price elasticities for both red and processed meat decreased by survey year, meaning they became less sensitive to price (S12 Table in S1 File). Stratified by income, the own-price elasticities of both red and processed meat decreased as income increased, meaning higher-income households were less sensitive to price increases in red and processed meat (Fig 1). In the lowest- and highest-income group, red meat was more elastic than processed meat, therefore, there were larger reductions in demand for red meat when price increased compared to processed meat for these groups. The own-price elasticity of red meat was more inelastic compared to processed meat in the middle-income quintiles.

An increase in the price of red meat was associated with an increase in the demand of F&V, grains, dairy, and legumes indicating these food groups were substitutes to red meat (Table 1). In contrast, price increases to red meat were associated with decreases in demand for all other food groups including discretionary foods, meaning that these food groups were complements to red meat. Over time, seafood, legumes, and processed meat became stronger complements, F&V a stronger substitute, and dairy and eggs weaker substitutes to red meat (S2 Text, S13 Table in S1 File).

An increase in the price of processed meat was associated with an increase in demand for legumes, seafood, poultry, and grains. Complements for processed meat were eggs, dairy, red meat, and discretionary foods. Over time, F&V became a complement instead of a substitute, red meat became a stronger substitute, and poultry was no longer a substitute to processed meat (S2 Text, S14 Table in S1 File). Looking at the food groups with the highest cross-price elasticities, F&V was a strong substitute to red meat and legumes was a strong substitute to processed meat, whereas discretionary foods were a strong complement to red meat and eggs were a strong complement to processed meat (Table 1).

Substitution and complement patterns varied by income group (S15 Table in S1 File). Comparing the highest to the lowest income group, discretionary foods, seafood, and processed meat were stronger complements to red meat. In the lowest income group, dairy, eggs, and F&V were stronger substitutes and poultry was a stronger complement to red meat.

**Table 1. Own- and cross-price elasticities of demand for red and processed meat and other food groups in Mexico, 2018-2022.**

| Food Category | Own-price elasticity | Cross-price elasticity with red meat | Cross-price elasticity with processed meat |
|---|---|---|---|
| Dairy | −1·54.(0.01)* | 0.08 (0.01)* | −0.11 (0.01)* |
| Eggs | −0.48 (0.04)* | 0.05 (0.03) | −0.20 (0.02)* |
| Discretionary Foods | −1.26 (0.00)* | −0.49 (0.01)* | −0.04 (0.00)* |
| Seafood | −2.09 (0.04)* | −0.41 (0.07)* | 0.34 (0.04)* |
| Fruits and Vegetables | −1.77 (0.01)* | 0.42 (0.01)* | 0.01 (0.01) |
| Grains, roots, and tubers | −1.10 (0.01)* | 0.32 (0.01) | 0.02 (0.01)* |
| Legumes, nuts, and seeds | −0.95 (0.04)* | 0.07 (0.05)* | 0.37 (0.03)* |
| Poultry | −0.65 (0.01)* | −0.29 (0.02)* | 0.04 (0.01)* |
| Processed Meat | −0.83 (0.04)* | −0.09 (0.04)* | – |
| Red Meat | −0.79 (0.04)* | – | −0.04 (0.01)* |

*Significant at 1%. Non-compensated price elasticity standard errors in parentheses.

The 'Other' food group category was omitted to avoid collinearity in OLS estimation.

Demand shifters included educational attainment of the head of household, year of survey (2018, 2020, and 2022), region of residence (urban vs rural), adult equivalent based on estimations derived from the Mexican population.

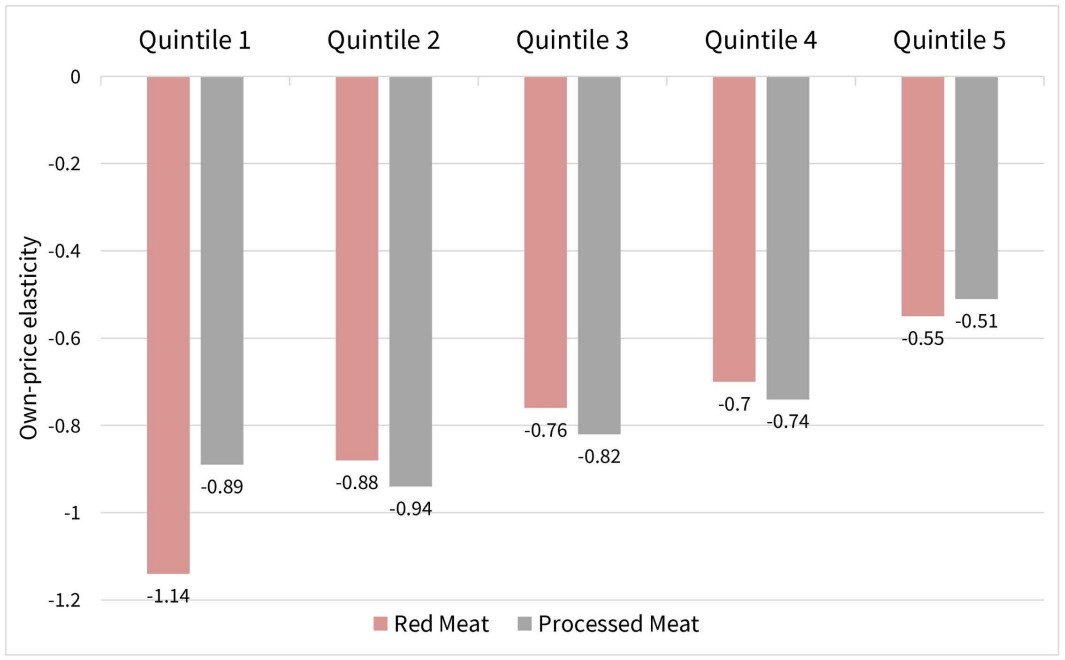

**Fig 1. Own-price elasticity of demand for red and processed meat by income quintile in Mexico, 2018-2022.** All estimations are significant at 1%. The 'Other' food group category was omitted to avoid collinearity in OLS estimation. Demand shifters included educational attainment of the head of household, year of survey (2018, 2020, and 2022), region of residence (urban vs rural), and adult equivalent based on estimations derived from the Mexican population.

For processed meat, discretionary foods and red meat were stronger complements and poultry a stronger substitute in the highest income group compared to the lowest. Moreover, F&V and seafood were stronger substitutes and dairy and eggs were stronger complements to processed meat in the lowest compared to the highest income group.

### Price elasticities of meat by cut quality

The own-price elasticities for cheaper and more expensive cuts of meat were −0.83 and −0.78, respectively (S2 Text; S16–S17 Tables in S1 File; S2 Fig in S1 File). Eggs and F&V were complements to cheaper cuts of meat whereas they were substitutes for expensive cuts of meat. Seafood and poultry were substitutes for cheaper cuts of meat but complements to expensive cuts. Grains and F&V were stronger substitutes to expensive cuts of meat compared to cheaper cuts and legumes were a stronger substitute to cheaper cuts. Stratified by income quintile, cheaper cuts were more price elastic than expensive cuts in the lowest income group whereas in the top income quintiles the opposite was true (S2 Fig in S1 File). Additionally, there were differential cross-price elasticities by income group (S17 Table in S1 File). Poultry and dairy were stronger substitutes and eggs, discretionary foods, grains, and expensive cuts of meat were stronger complements to cheaper cuts of meat in the highest compared to lowest income group. Whereas, eggs and grains were stronger substitutes, and dairy, discretionary foods and cheaper cuts of meat were stronger complements to expensive cuts of meat in highest compared to lowest income group.

Cheaper cuts of meat tended to be processed meat whereas more expensive cuts of meat tended to be red meat (hence, the similarity between own-price elasticities) (S4 Table in S1 File). Therefore, we did not simulate demand responses to price increases by cut of meat.

## Price elasticities estimated with Seemingly Unrelated Regression

The own-price elasticities estimated using Seemingly Unrelated Regression in the sensitivity analysis were broadly consistent for red and processed meat, although slightly more inelastic than those found in our primary analysis (−0.61 and −0.75, respectively; S18 Table in S1 File). Own-price elasticities were similar for the remaining the food groups modeled in the system. Cross-price elasticities were mostly consistent between both estimation methods; however, slight differences were observed (S18 Table in S1 File). In the sensitivity analysis, dairy and legumes were complements to red meat whereas they were weak substitutes in the primary analysis. Moreover, for processed meat, no relationship was found between the price of processed meat and demand for discretionary compared to a weak complement in the primary analysis.

## Applying tax scenarios

Nearly half (45%) of Mexican adults consumed red meat and 32% consumed processed meat on a given day in 2016 (S2 Text; S19 Table in S1 File).

When a price increase of 30% was applied to both red and processed meat (Scenario 1), red meat consumption decreased to levels near the Mexican Dietary Guidelines in the general adult population (34g/day vs. 30g/day recommended), but processed meat consumption did not (16g/day vs 0g/day recommended) (Table 2). Daily intake of animal-source protein foods increased from 132g/day to 145g/day vs. 140g/day and 84g/day recommended by the Mexican Dietary Guidelines and EAT-*Lancet*, respectively. The lowest income consumers came closer to red and processed meat dietary targets than the highest income consumers (Table 2). Red and processed meat consumers did not reduce demand for meat as much as the general adult population (Table 2; Figs 2 and 3). Scenario 1 led to the largest declines in percent demanded of discretionary foods, poultry, and red meat and the largest increases in demand for F&V and grains (Fig 2). For red meat and processed consumers, it resulted in the largest increases in F&V and grains and the largest decreases in discretionary foods, seafood, and red meat (Fig 3). Scenario 1 increased F&V consumption in the lowest income group and reduced discretionary foods in the highest income group (Fig 4).

After a 15% price increase for red meat and 30% for processed meat (Scenario 2), demand for red meat slightly decreased as red meat is a complement (Table 2; Fig 2). Scenario 3, which comprised of a 7% price increase to red meat and 47% price increase to processed meat, achieved levels of red and processed meat consumption aligned with the Mexican Dietary Guidelines in the lowest income group whereas among red meat consumers it resulted in minimal changes in demand of red meat (Table 2; Figs 3 and 4).

A 100% price increase to processed meat only (Scenario 4) aligned red and processed consumption closely to the Mexican Dietary Guidelines, resulting in the largest declines in processed meat across all subgroups compared to other scenarios (Table 2; Figs 2–4). For the general adult, red meat consumer, and lowest income population, demand for processed meat reduced by approximately 84% with slight reductions in red meat due to their complementarity. This scenario resulted in the greatest reductions in processed meat among processed meat consumers and moderately reduced demand for red meat since red meat is a stronger complement to processed meat in this group compared to the reverse in red meat consumers (Fig 3). Scenario 4 led to the largest declines in dairy and eggs but largest increases in seafood, legumes, and poultry in the adult population (Fig 2). It also achieved the greatest consumption of animal-source protein foods in the lowest income group (Table 2). For red meat consumers, it led to the largest increases in seafood and legumes whereas among processed meat consumers it led to the largest increases in legumes and eggs (Fig 3). In the lowest income group, it resulted in greater increases in demand for seafood and F&V whereas in the highest income group it led to greater increases in demand for legumes and poultry (Fig 4).

Lastly, since no predefined price increase scenarios reached the processed meat 0g/day target, we calculated that an approximate 120% price increase would be required for processed meat to achieve such a target in the adult Mexican population.

**Table 2. Estimated percent change in quantity demanded and average daily consumption (g) by population according to price increases to red and processed meat in Mexico.**

| | Red Meat | | Processed Meat | | Animal-source protein foods |
|---|---|---|---|---|---|
| | Reduction in quantity demanded, % | Consumption after price increase (g/day) | Reduction in quantity demanded, % | Consumption after price increase (g/day) | Consumption after price increase (g/day) |
| **Reference: no price increase (baseline consumption)** | | | | | |
| **Adult Population ≥18 years of age** | 0 | 45.8 (4.9) | 0 | 22.4 (3.1) | 132.3 (6.9) |
| **Consumption Status** | | | | | |
| Red Meat Consumers | 0 | 101.2 (9.6) | 0 | 16.0 (2.6) | 162.4 (12.8) |
| Processed Meat Consumers | 0 | 43.0 (11.8) | 0 | 70.6 (7.2) | 121.0 (11.0) |
| **Income groups** | | | | | |
| Lowest | 0 | 26.3 (5.4) | 0 | 11.2 (4.5) | 100.2 (12.6) |
| Highest | 0 | 56.2 (9.7) | 0 | 28.2 (5.9) | 144.1 (12.4) |
| **Scenario 1–30% price increase both red and processed meat** | | | | | |
| **Adult Population ≥18 years of age** | −24.7% | 34.5 | −27.8% | 16.2 | 144.9 |
| **Consumption Status** | | | | | |
| Red Meat Consumers | −19.5% | 81.4 | −27.8% | 11.6 | 161.5 |
| Processed Meat Consumers | −25.4% | 32.1 | −19.2% | 57.1 | 130.8 |
| **Income groups** | | | | | |
| Lowest | −35.4% | 17.0 | −28.9% | 8.0 | 132.4 |
| Highest | −18.7% | 45.7 | −22.8% | 21.8 | 153.8 |
| **Scenario 2–15% price increase red meat and 30% price increase processed meat** | | | | | |
| **Adult Population ≥18 years of age** | −12·9% | 39.9 | −26.4% | 16.4 | 152.3 |
| **Consumption Status** | | | | | |
| Red Meat Consumers | −9·8% | 91.3 | −26.8% | 11.7 | 174.2 |
| Processed Meat Consumers | −17·5% | 36.7 | −15.4% | 58.2 | 136.7 |
| **Income quintiles** | | | | | |
| Lowest | −18·3% | 21.5 | −27.7% | 8.1 | 138.4 |
| Highest | −10·5% | 50.3 | −19.1% | 22.8 | 160.7 |
| **Scenario 3–7% price increase red meat and 47% price increase processed meat** | | | | | |
| **Adult Population ≥18 years of age** | −7.3% | 42.5 | −39.9% | 13.5 | 157.4 |
| **Consumption Status** | | | | | |
| Red Meat Consumers | −4.5% | 96.6 | −40.9% | 9.5 | 182.1 |
| Processed Meat Consumers | −13·1% | 37.4 | −25.6% | 52.5 | 139.7 |
| **Income groups** | | | | | |
| Lowest | −9·8% | 23.7 | −42.0% | 6.7 | 142.0 |
| Highest | −7·5% | 52.0 | −25.8% | 21.0 | 165.1 |
| **Scenario 4–100% price increase to processed meat only** | | | | | |
| **Adult Population ≥18 years of age** | −3.8% | 44.0 | −83.5% | 3.7 | 164.4 |
| **Consumption Status** | | | | | |
| Red Meat Consumers | −0.05% | 101.2 | −86.0% | 2.2 | 191.7 |
| Processed Meat Consumers | −18.1% | 35.2 | −52.8% | 33.4 | 141.7 |

*(Continued)*

**Table 2.** (Continued)

| | Red Meat | | Processed Meat | | Animal-source protein foods |
| --- | --- | --- | --- | --- | --- |
| | Reduction in quantity demanded, % | Consumption after price increase (g/day) | Reduction in quantity demanded, % | Consumption after price increase (g/day) | Consumption after price increase (g/day) |
| **Income groups** | | | | | |
| Lowest | −3.7% | 25.3 | −88.3% | 1.3 | 146.0 |
| Highest | −7.9% | 51.7 | −51.2% | 13.8 | 170.8 |

Values in reference scenario are survey-weighted mean (SE) grams per day consumed per capita. Animal-source protein foods refer to the sum of red meat, fish, poultry, and eggs consumed per day. Animal-source protein food daily intake was determined as the sum of red meat, poultry, eggs, and fish as described in S1 Table in S1 File.

The average grams/day were calculated applying the percentage reduction in demand to actual single-day weighted intake for each population. For animal-source protein foods, the percent changes in demand were applied to baseline consumption of each respective food group and then summed to obtain estimated daily intake after price increase.

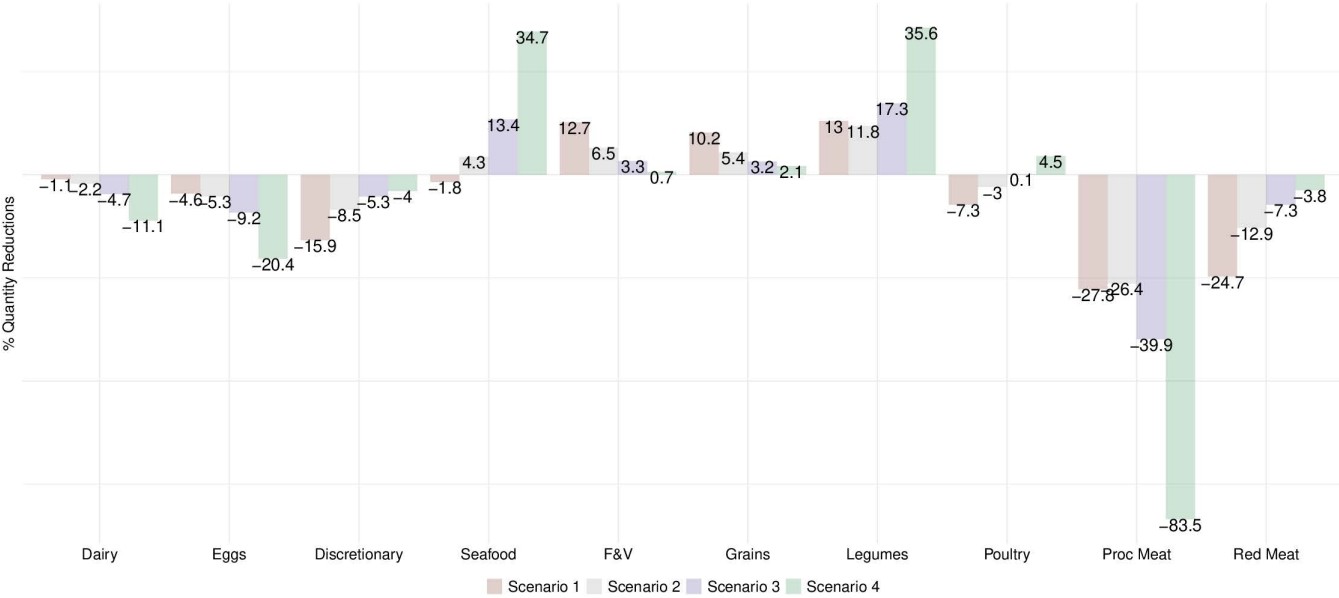

**Fig 2. Estimated percent change in quantity demanded according to price increases in red meat and processed meat in Mexican Adults, 2018-2022.** Values are percent change in quantity demanded of food groups according to four price increase scenarios for red and processed meat estimated by own- and cross-price elasticities. Scenario 1: 30% price increase to both red and processed meat; Scenario 2: a 15% price increase to red meat and a 30% price increase to processed meat; Scenario 3: a 7% price increase to red meat and a 47% price increase to processed meat; Scenario 4: 100% price increase to processed meat only. F&V refers to 'fruits and vegetables', Grains refers to 'Grains, roots, and tubers', Legumes refers to 'legumes, nuts, and seeds', and Proc Meat refers to processed meat.

## Dietary substitutions over time in MTC

Red and processed meat were also complements to each other in a longitudinal study of Mexican women (S20 Table in S1 File). Changes in red, processed, and total meat intake from 2006–2008 and 2008–2014 were negatively correlated with changes in F&V, supporting evidence that this food group is a substitute for red and total meat (S20 Table in S1 File). Changes in processed and total meat intake were positively correlated with changes in poultry, supporting this food group

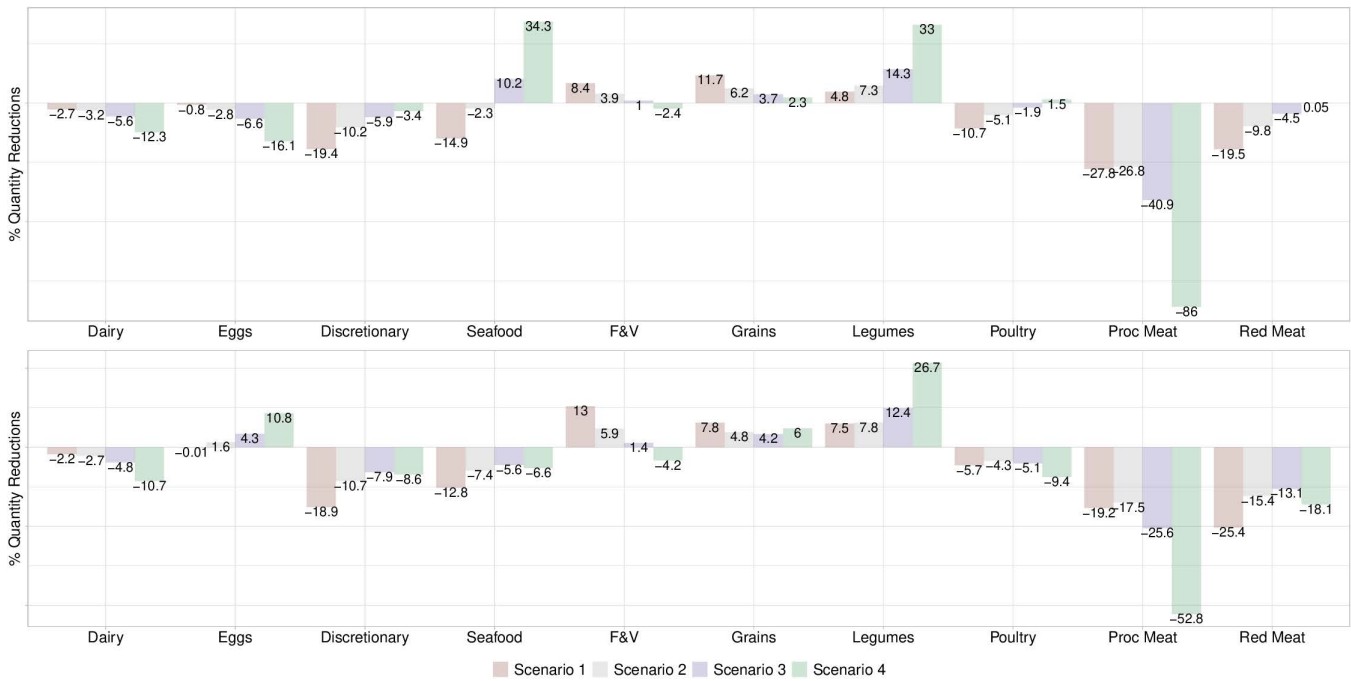

**Fig 3. Estimated percent change in quantity demanded according to price increases in red meat and processed meat among consumers in Mexico, 2018-2022. Top graph** Values are percent change in quantity demanded of food groups for red meat consumers according to four price increase scenarios for red and processed meat based on own- and cross-price elasticities calculated for households who reported non-zero red meat consumption. **Scenario 1**: 30% price increase to both red and processed meat; **Scenario 2:** a 15% price increase to red meat and a 30% price increase to processed meat; **Scenario 3:** a 7% price increase to red meat and a 47% price increase to processed meat; **Scenario 4:** 100% price increase to processed meat. Bottom graph: Values are percentage change in quantity demanded of food groups for processed meat consumers according to four price increase scenarios for red and processed meat based on own- and cross-price elasticities calculated for households who reported non-zero processed meat consumption. **Scenario 1**: 30% price increase to both red and processed meat; **Scenario 2:** a 15% price increase to red meat and a 30% price increase to processed meat; **Scenario 3:** a 7% price increase to red meat and a 47% price increase to processed meat; **Scenario 4:** 100% price increase to processed meat only. F&V refers to 'fruits and vegetables', Grains refers to 'Grains, roots, and tubers', Legumes refers to 'legumes, nuts, and seeds', and Proc Meat refers to processed meat.

as a complement. Moreover, changes in seafood intake were positively correlated with processed and total meat intake in both periods suggesting this food group is a complement. Lastly, we did not find correlations between changes of red, processed, or total meat intake with discretionary foods, dairy, nor grains.

## Discussion

Moderate price increases to meat in Mexico could help the general adult population achieve the Mexican Dietary Guidelines' target for red meat, but more substantial increases in the price of processed meat (about 120%) are required to achieve the target of 0g/day. The own-price elasticities for red and processed meat are relatively similar; therefore, increasing the price by the same ratio leads to similar reductions in demand for each. However, it results in different changes in demand for other food groups important for nutrition and health [14]. As red meat consumption goes down, consumption of F&V, grains, and dairy is likely to go up and consumption of discretionary foods, poultry, seafood, and processed meat is likely to go down. As processed meat consumption goes down, legumes, seafood, poultry, and grains are likely to go up and consumption of dairy, eggs, discretionary foods, and red meat is likely to go down. Increases in the price of red and processed meat increased consumption of nutrient- and protein-rich foods (legumes, seafood, and poultry) in the lowest income group. The Mexican Dietary Guidelines recommend replacing red and processed meat with

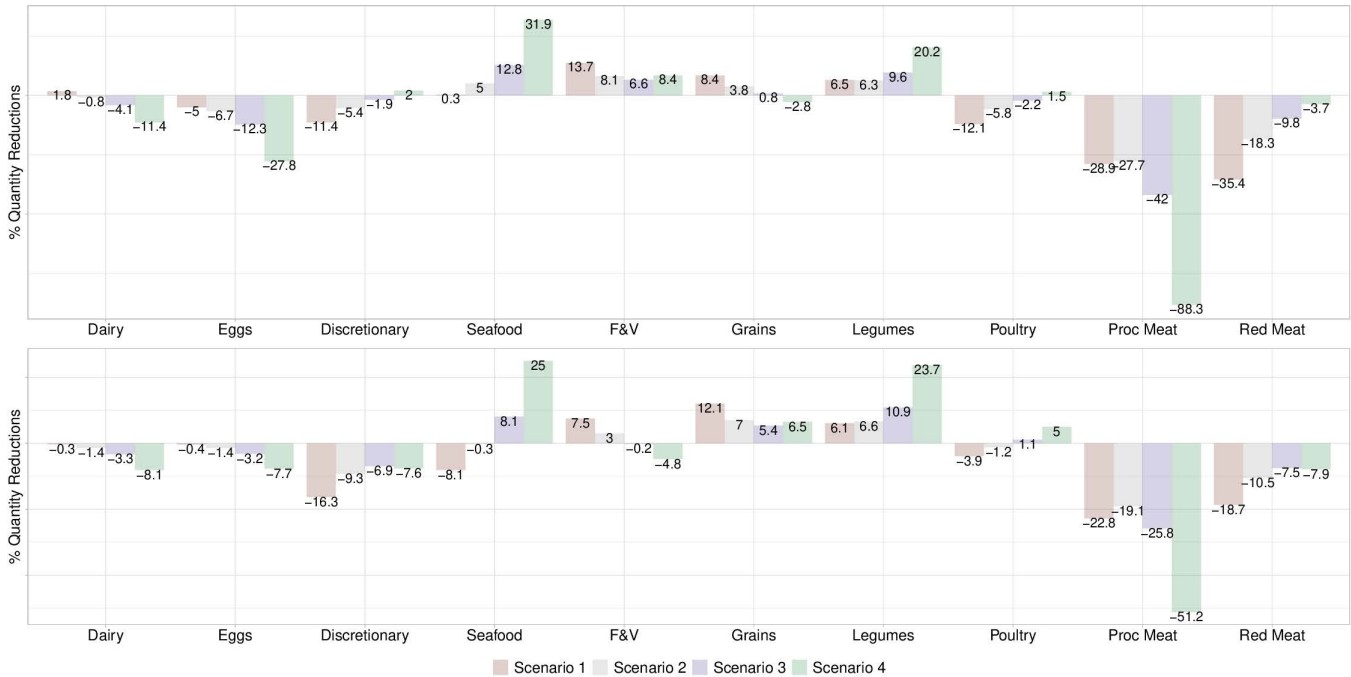

**Fig 4. Estimated percent change in quantity demanded according to price increases in red meat and processed meat in lowest and highest income groups in Mexico, 2018-2022. Top graph** Values are percent change in quantity demanded of food groups for low-income consumers according to four price increase scenarios for red and processed meat based on own- and cross-price elasticities calculated for households in lowest income quintile. **Scenario 1**: 30% price increase to both red and processed meat; **Scenario 2:** a 15% price increase to red meat and a 30% price increase to processed meat; **Scenario 3:** a 7% price increase to red meat and a 47% price increase to processed meat; **Scenario 4:** 100% price increase to processed meat only. Bottom graph: Values are percentage change in quantity demanded of food groups for high income consumers according to three price increase scenarios for red and processed meat based on own- and cross-price elasticities calculated for households in highest income quintile. **Scenario 1**: 30% price increase to both red and processed meat; **Scenario 2:** a 15% price increase to red meat and a 30% price increase to processed meat; **Scenario 3:** a 7% price increase to red meat and a 47% price increase to processed meat; **Scenario 4:** 100% price increase to processed meat only. F&V refers to 'fruits and vegetables', Grains refers to 'Grains, roots, and tubers', Legumes refers to 'legumes, nuts, and seeds', and Proc Meat refers to processed meat.

beans, lentils, egg, poultry, and fish [14]. Therefore, increasing the price of red and processed meat may promote greater adherence to the guidelines, especially in lower income groups, which has been designed to deliver a micronutrient adequate diet, and therefore, may improve diet quality due to increases in consumption of nutrient-dense foods and reductions in discretionary foods.

We found that a substantial price increase for processed meat alone (i.e., a 100% increase) may be a better strategy than moderately increasing the price of both red and processed meat. Processed meat poses a greater health burden [35], and has been consistently found to require higher pricing than red meat to internalize associated health costs [17]. This scenario presented the highest increases in protein-rich foods such as seafood, legumes, and poultry out of the four scenarios in all groups and also increased F&V. Furthermore, it maintained consumption of red meat at recommended levels in the lowest income group, for whom moderate consumption may be important to ensure sufficient intake of key nutrients [23,36].

Our findings are in agreement within the context of the relevant literature. LA/AIDs was previously carried out using ENIGH 2002−2012 in Mexico for urban households only and derived an own-price elasticity of approximately −0.803 for meats (including red, processed meats, and poultry) [37]. In this study, the own-price elasticity for seafood was also the highest among the food groups as it was in our findings, further corroborated in another study comparing pre- and

during COVID-19 food demand in Mexico (−2.07 for seafood before COVID-19) [38]. A meta-analysis of global own-price elasticities for meat from 2010 obtained a median own-price elasticity for meat (including ruminant, non-ruminant, poultry, and fish) of −0.762 [39]. More specifically, a pooled own-price elasticity of −0.87 was obtained for beef and −0.65 for poultry [39]. Our own-price elasticities are roughly consistent with these estimates despite difficulty in comparison due to heterogeneity in food groupings and context-specific demand systems. We found that red and processed meat became more inelastic over time (especially after 2020). Shifts in household purchasing patterns due to associated reductions in employment and income following the COVID-19 pandemic could have played a role. An analysis evaluating changes in household purchases from 2018 to 2020 in ENIGH found an increase in basic food purchases: specifically, an increase in processed meat purchases (by approximately 18%) in the lowest income quintile and an increase in SSB purchases (grouped with discretionary foods in our analysis and a complement to red and processed meat) in the rural population in Mexico [40]. Although, the proportion of household income allocated to food did not change between these two time periods.

Previous studies have proposed taxes on meat to internalize negative externalities associated with high meat consumption, be it environmental consequences, human health, animal welfare, or social factors including labor [41]. In Sweden, France, and the Netherlands, price increases ranging between 15% to 34% were modeled for beef, all animal products, and total meat (including red, white, and processed) [19,20,42] and in Brazil these ranged from 5% to 60% for beef to account for environmental and health costs [21]. A global modeling study found that the price of red meat should increase on average by 21.26% for high-income and 6.51% for upper-middle income countries to account for associated health costs [17]. Processed meat required greater price increases due to its greater health burden: 111% for high-income countries (up to 200% in the United States) and 46.85% for upper middle-income countries [17]. In the real world, meat taxes have mostly been considered in Europe, but they are also being implemented and discussed in Latin America [21,43,44]. For example, Colombia recently passed a tax on ultra-processed food that includes some processed meats, 'embutidos' [43]. Brazil is currently proposing a health- and sustainability-motivated tax to ultra-processed foods which would include some processed meats [44].

Our study taken together with the cumulative evidence suggest that large price increases are needed to achieve sufficient reductions in meat consumption, under a critical timeline if we want to meet urgent climate goals such as <2 °C degrees of warming [8] while addressing diet-related health burdens [45]. As noted, imposing a tax only on processed meat may be more politically feasible given its stronger association with chronic disease, particularly colorectal cancer and diabetes [9,10,33,35], and that moderate consumption of red meat may still benefit lower income, indigenous populations as a source of nutrients, such as heme iron and vitamin B12 [23]. To date, we are only aware of a beverage-specific tax that has been implemented at rates similar to those proposed for processed meat in this study, namely in Saudia Arabia and the United Arab Emirates [46]. The food tax in Colombia is expected to increase to 20% by 2025 making it one of the highest food tax rates, globally [43]. Factors such as public acceptance, feasibility, and implementation are important. Ensuring that dietary substitutes are available and affordable are critical for feasibility. Furthermore, several conditions may favor such a policy. Mexico currently has political momentum to pass progressive food legislation in favor of health and sustainability. Beyond incorporating sustainability into the national dietary guidelines as a policy tool to justify enforcement, Mexico implemented front-of-package labeling with nutrient warnings on packaged foods and beverages in 2020 [47], and has recently passed a ban on the sale of ultraprocessed foods and beverages in schools to be implemented in March 2025 [48]. Additionally, Mexico enacted a General Law on Adequate and Sustainable Nutrition in 2024 that constitutionally protects the right to adequate, nutritious, and sustainably produced food [49], which may serve as a normative framework for complementary legislation in the future. The high suggested price increases in this study are warranted considering the concurrent environmental and health crises, as well as the Mexican government's present political willingness to act on these issues. These findings serve as an ambitious scenario to understanding the tax rates needed to achieve sufficient changes in consumption across the Mexican population to meet health and environmental objectives.

Ambitious targets may serve as a bargaining tool or escalate urgency, given that in practice, negotiation with government, industry, and the public often result in tax rates or policy measures that can fall short of what is deemed necessary by evidence [1,50,51].

We found that lower income groups were more responsive to changes in prices of red and processed meat. Moreover, in this case, lower income groups consumed less red and processed meat and spent a lower proportion of their food budget on these foods. Given that lower income groups allocate a smaller proportion of their food budget to meat – despite being more price sensitive – the tax would not necessarily impose a disproportionate burden on them. Moreover, lower income groups tend to have greater adherence to the sustainable and healthy traditional Mexican Diet, which serves as a foundation for Mexican National Dietary Guidelines [23,26]. Therefore, red and processed meat consumption in this group would change by a smaller proportion, and food substitutions would align more closely with traditional dietary patterns as was observed in our models [26]. In the case of health-motivated taxes, reductions in consumption among lower income groups can actually mitigate health disparities and healthcare expenditures [52]. On average, adherence to the 2023 Mexican Dietary Guidelines is less expensive than the current diet (by 21%) in the general population since costs are offset by red meat consumption. However, the 2023 Mexican Dietary Guidelines diet was more expensive than the current diet among lower income groups mostly due to an increase in expenditure on nuts, fruits, and vegetables [26]. Therefore, special consideration should be taken to ensure the affordability of sustainable and healthy diets, including dietary substitutes, in this population. Again, taxing only processed meat does not pose a risk to diet quality, as its associated health risks outweigh potential benefits. Therefore, this approach may also help mitigate concerns regarding affordability of a 'healthy' diet among lower income populations. Nutrition surveillance of at-risk populations is critical to inform how compensatory measures may be designed to enhance dietary quality. Tax revenue recycling in the form of targeted lump sum transfers can be used to benefit lower income groups and counteract any potential regressivity of the tax [53], subject to Congress and government approval as, in Mexico, revenues are not earmarked. They can be used to subsidize the cost via subsidies, food vouchers, or cash transfers to enable purchasing of healthier, more nutrient-rich foods such as F&V and legumes, especially since both food groups were stronger substitutes to red and processed meat in the lowest compared to highest income group. This was explored in a recent study evaluating the regressivity of meat taxes across Europe [53]. It found that tax recycling through lump sum transfers to the lowest quintile were more equitable than lowering the price of fruits and vegetables [53]. Moreover, a social-cost benefit analysis applied price elasticities obtained from a literature review to test price increases of 15 and 30% to total meat (including red, white and processed) and found that a 10% decrease in price of fruit and vegetables counteracted regressivity among lower income groups [19]. Food vouchers may facilitate more direct targeting of subgroups most at-risk of nutrient inadequacy and be designed for the purchase of nutrient-rich foods, particularly rich in iron and vitamin B12, such as seafood, poultry, or eggs. In a randomized study conducted in Ecuador, food vouchers were found to increase dietary diversity as opposed to in-kind food or cash assistance [54]. In addition to financial incentives, nutritional education may be a complementary strategy to ensure that these populations are making the 'healthier' choice.

This analysis has many strengths in that it estimated the Mexican food demand system pooling three survey rounds of nationally representative household expenditure data to estimate price elasticities of key food groups. Only two previous studies [20,42] that modeled meat taxes calculated price elasticities based on the country-specific food demand systems as the other papers either used price elasticities from literature reviews [19] or calculated the marginal costs of consumption to health [17]. Additionally, we were able to evaluate demand responses to price in the Mexican population as well as by income and meat consumption status and apply these scenarios to actual red and processed meat intake based on dietary data. Lastly, we further confirmed these findings in an independent analysis of longitudinal dietary behaviors.

However, these findings should be interpreted with consideration of their limitations. First, price elasticities were modeled using LA/AIDs meaning that the relationship between price and demand was assumed to be linear. Future work may seek to determine whether there is non-linearity in the relationship meaning price increases would result in non-linear

responses in demand. Second, we excluded food expenditures for consumption away from the home since information on specific foods purchased was not collected in ENIGH (comprising approximately 16% of household food budgets across all three ENIGH survey rounds) especially among higher income households [55] which may have introduced bias. Third, we acknowledge that we cannot rule out potential bias in the model that may have influenced the relationship between price and demand. However, we were able to account for multiple potential sources of endogeneity such as price outliers and demand shifters that may influence the relationship between price and quantity demanded in addition to calculating unit values at the municipality level to reduce this risk. Fourth, it should be noted that we are only modeling demand responses to price increases and not supplier responses. It cannot be assumed that declines in consumption will result in declines in production as Mexican livestock producers may choose to export instead of sell domestically, which may result in 'carbon leakage' [56]. Nonetheless, beef is an international food commodity so no matter where it is produced, reducing its consumption may mitigate its contribution to environmental change.

## Recommendations for future work

Future studies should examine the potential to explore combined policy mechanisms such as front-of-package labeling or signaling (i.e., public framing and messaging) in combination with taxes given combined policy mechanisms have been found to be more effective in trial settings since price is not the only determinant of food choice [2,18,57]. Investigation into social determinants of red and processed meat consumption could enhance the effectiveness of food tax policies. For instance, knowledge of within-country cultural differences could be leveraged to tailor targeted policies. Red meat consumption, for example, is higher in Northern and urban regions of the country [26] where it is more prominently featured in staple dishes and tied to culinary identity [58]. However, there is reframing of this narrative in efforts seeking to revive the traditional Mexican diet – such as the 2023 Mexican Dietary Guidelines – as the popularity of traditional foods like beans has declined, while meat consumption has increased, particularly after the enactment of the North American Free Trade Agreement in 1994 [59,60]. Expanding on these complementary public relations campaigns – such as those led by the Center for Research on Nutrition and Health in Mexico – could highlight food substitutes that are culturally tailored to regional dishes as well as the nutritional and environmental implications to encourage these dietary swaps. Moreso, research on whether lower income families will purchase and cook dietary substitutes is essential in addition to better understanding the nutritional implications of such reductions, ensuring that substitution patterns adequately deliver energy and nutrient requirements. Moreover, further research into the nutrient implications resulting from reductions in red and processed meat – especially for iron and vitamin B12 – while considering substitutions in the rest of the diet is a key priority, especially among lower income populations. Adherence to the Mexican Dietary Guidelines and a diverse diet is indicative of micronutrient adequacy, not just red and processed meat dietary targets, therefore, this should be considered in future micronutrient evaluations. Additionally, a political analysis examining the key stakeholders and political instruments that would facilitate the passing of such legislation – as was done for the 2014 tax in Mexico [51] – would be beneficial considering the large reductions in demand required to meet sustainable and healthy dietary targets. Lastly, research should estimate the impact of these taxes on environmental outcomes.

## Concluding remarks

This study provides evidence required for the design of a tax which may be used to inform policymakers in Mexico. Increasing the price of red and processed meat in Mexico may be effective in reducing consumption to address both health- and environment- related challenges. Specifically, substantially increasing the price of processed meat may be the best option for maximum health gains, and little risk to diet quality, while reducing environmental costs. Lower income groups are more sensitive to price increases in red and processed meat, but there is evidence they are substituting these foods with other protein- and nutrient-rich foods such as seafood, poultry, and legumes. Moreover, a tax may be combined with tax recycling or lump sum transfers to make these options more affordable to lower income groups.

Government-supported price increases may be an effective way to shifting consumption to more sustainable and healthy diets.

## Supporting information

**S1 File. Supplemental tables and figures.**
(DOCX)

**S1 Text. Supplemental methods.**
(DOCX)

**S2 Text. Supplemental results.**
(DOCX)

**S3 Text. Supplemental references.**
(DOCX)

## Acknowledgments

We would like to thank Arne Heningsen, author of the micEconAids package in R, for guidance on its implementation and answering our queries.

## Author contributions

**Conceptualization:** Kaela Connors, Juan A. Rivera, Peter Alexander, Lindsay M. Jaacks, M. Arantxa Colchero.

**Data curation:** Carolina Batis, Dalia Stern, Martín Lajous.

**Formal analysis:** Kaela Connors, M. Arantxa Colchero.

**Funding acquisition:** Kaela Connors.

**Investigation:** Kaela Connors, Juan A. Rivera, Carolina Batis, M. Arantxa Colchero.

**Methodology:** Kaela Connors, Peter Alexander, Lindsay M. Jaacks, Dalia Stern, M. Arantxa Colchero.

**Supervision:** Juan A. Rivera, Peter Alexander, Lindsay M. Jaacks, Carolina Batis, Dalia Stern, Martín Lajous, M. Arantxa Colchero.

**Validation:** Kaela Connors, M. Arantxa Colchero.

**Writing – original draft:** Kaela Connors.

**Writing – review & editing:** Kaela Connors, Juan A. Rivera, Peter Alexander, Lindsay M. Jaacks, Carolina Batis, Dalia Stern, Martín Lajous, M. Arantxa Colchero.

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
