## [Decision Letter · Decision Letter 0]

Dear Dr. Connors,

Thank you for submitting your manuscript to PLOS ONE. After careful consideration, we feel that it has merit but does not fully meet PLOS ONE’s publication criteria as it currently stands. Therefore, we invite you to submit a revised version of the manuscript that addresses the points raised during the review process.

**ACADEMIC EDITOR:**

We look forward to receiving your revised manuscript.

Kind regards,

Charles Odilichukwu R. Okpala, PhD

Academic Editor

PLOS ONE

Journal Requirements:

“Kaela Connors received financial support from the Edinburgh Earth, Ecology, and Environment Doctoral Training Partnership”

Additional Editor Comments:

A wide range of feedback for major revision, authors should improve the work

Reviewers' comments:

Reviewer's Responses to Questions

**Comments to the Author**

1. Is the manuscript technically sound, and do the data support the conclusions?

Reviewer #1: Yes

Reviewer #2: Yes

Reviewer #3: Yes

Reviewer #4: Partly

2. Has the statistical analysis been performed appropriately and rigorously?

Reviewer #1: I Don't Know

Reviewer #2: Yes

Reviewer #3: Yes

Reviewer #4: N/A

3. Have the authors made all data underlying the findings in their manuscript fully available?

Reviewer #1: Yes

Reviewer #2: Yes

Reviewer #3: Yes

Reviewer #4: Yes

4. Is the manuscript presented in an intelligible fashion and written in standard English?

Reviewer #1: Yes

Reviewer #2: Yes

Reviewer #3: Yes

Reviewer #4: No

Reviewer #1: First, I would like to commend the authors on a well-conducted and thorough analysis of the effects of price elasticity on the demand for red and processed meat in Mexico. The manuscript provides a detailed exploration of the interactions between price changes and consumer behavior, which is highly relevant for informing public health policies aimed at reducing meat consumption. The consideration of different price increase scenarios effectively highlights the complexity of influencing dietary behaviors through economic interventions, and modeling these effects on a national scale is an important contribution to the literature on food policy and public health.

The manuscript is technically sound in many aspects, with a well-structured methodology and robust statistical analysis. However, there are areas that could benefit from further refinement. Specifically, the interpretation of demand elasticity changes over time would be enhanced by a more in-depth discussion of the socio-economic factors that might influence these trends. While the conclusions are generally well-supported by the data, the practical feasibility of the substantial price increases proposed—especially to achieve dietary targets—could be discussed further to address potential consumer responses and implementation challenges.

The manuscript largely adheres to reporting standards, providing transparency in data sources and methodologies. However, while the statistical significance of the findings is frequently noted, there is insufficient discussion of their practical implications. Expanding on how these statistically significant results might translate into real-world outcomes would greatly enhance the reader's understanding of their relevance and potential impact.

Reviewer #2: The manuscript needs further clarification and enhancement in some key areas. For example, the methodology, especially the estimation of price elasticities, would be enhanced by a detailed discussion of assumptions made during the analysis. This might include suggestions for investigating possible confounding variables-what affects consumer behavior other than price changes, such as cultural considerations or alternative sources for food products. This could also involve sensitivity analyses that prove the strength of the results found.

Secondly, there should be more discussion on implications related to the most vulnerable populations and increases in meat prices. While it is mentioned that consumers of meat from a low-income background are more responsive to any changes in price, some indication should be made concerning how this population can be supported in making healthier choices, given rising prices. This needs to be in regard to certain policy measures or even subsidies in terms of healthier alternatives. This, in turn, would allow a more detailed exploration of how meat taxation could impact both public health and equity.

Reviewer #3: First of all I would like to express my thankful for the editor of EJVS for their trust in reviewing this manuscript. This article is very interesting in the field of food safety in Mexico.

The introduction covers the problem, statement, and objectives of the study.

Material and methods explain in detail how authors collected and analyzed their data.

Results and discussion section are clearly and reflect the most abundant findings

Note: change all table and figure titles to understand more, change figure resolution

According to our opening we recommend Minor revision.

Reviewer #4: This is an economic report not a scientific paper

We assessed how to achieve sustainable, equitable, and healthy dietary targets for red and processed meat consumption in Mexico. The four specific aims were to (1) estimate own- and cross-price elasticities of red and processed meat, (2) estimate demand responses to price increases in red and processed meat under various scenarios and apply them to consumption data, (3) explore if price elasticities varied by household income and other effect modifiers (i.e., survey year, and consumption status) and (4) corroborate cross-sectional findings by evaluating dietary substitutions and complements to red and processed meat in a longitudinal cohort of Mexican women.

REJECT

**Do you want your identity to be public for this peer review?** For information about this choice, including consent withdrawal, please see our Privacy Policy

Reviewer #1: No

Reviewer #2: No

Reviewer #3: **Yes: ** wedfwefefefwefefwe

Reviewer #4: No

---

## [Author Response · Author response to Decision Letter 1]

13 Feb 2025

Thank you for your consideration. We have included point-by-point responses to the reviewers' comments in a rebuttal letter and have updated our funding statement and data availability statement for the third-party Mexican Teachers' Cohort in the Cover Letter to the Academic Editor as requested. We have additionally ensured that formatting complies with PLOS ONE requirements and have formatted our figures to comply with style requirements using the suggested online tool. We greatly appreciate the oppurtunity to respond to comments, and look forward to hearing back. We have pasted the responses below for your convenience.

Manuscript Title: Promoting adherence to the sustainable Mexican Dietary Guidelines through a tax on red and processed meat

We greatly appreciate the careful and thorough consideration of our manuscript. We have incorporated changes as per reviewer comments, which we believe have strengthened the manuscript. Please find our responses to each point below. Note that line numbers refer to those in the manuscript version without tracked changes.

Reviewer #1

First, I would like to commend the authors on a well-conducted and thorough analysis of the effects of price elasticity on the demand for red and processed meat in Mexico. The manuscript provides a detailed exploration of the interactions between price changes and consumer behavior, which is highly relevant for informing public health policies aimed at reducing meat consumption. The consideration of different price increase scenarios effectively highlights the complexity of influencing dietary behaviors through economic interventions, and modeling these effects on a national scale is an important contribution to the literature on food policy and public health.

Thank you for your careful review of our manuscript, and for highlighting the strengths of this analysis. Below we have addressed suggested changes in-line.

The manuscript is technically sound in many aspects, with a well-structured methodology and robust statistical analysis. However, there are areas that could benefit from further refinement.

1. Specifically, the interpretation of demand elasticity changes over time would be enhanced by a more in-depth discussion of the socio-economic factors that might influence these trends.

Response: We agree with the reviewer that there should be we expanded discussion on how socioeconomic factors could explain changes in price elasticities over time [2018, 2020, 2022]. Please find this included in Lines 365 - 372:

‘We found that red and processed meat became more inelastic over time (especially after 2020). Shifts in household purchasing patterns due to associated reductions in employment and income following the COVID-19 pandemic could have played a role. An analysis evaluating changes in household purchases from 2018 to 2020 in ENIGH found an increase in basic food purchases: specifically, an increase in processed meat purchases (by approximately 18%) in the lowest income quintile and an increase in SSB purchases (grouped with discretionary foods in our analysis and a complement to red and processed meat) in the rural population in Mexico (33). Although, the proportion of household income allocated to food did not change between these two time periods.’

Relatedly, we have included additional detail on socioeconomic factors that may influence the relationship between price and demand of food groups. As one approach, we accounted for this in our modeling of the Mexican food demand system by adjusting for demand shifters – variables that may influence the relationship between price and demand – in our estimation of uncompensated price elasticities of demand.

The the motivation for accounting for demand shifters in the demand system is now more clearly explained in Lines 105- 109:

‘We accounted for potential sociodemographic factors that could influence the relationship between price and demand through accounting for demand shifters in the model. These included educational attainment of the head of household, year of survey (2018, 2020, and 2022), region of residence (urban vs rural), and adult equivalent based on estimations derived from the Mexican population considering age and size composition of the household (23).’

We also stratified by socioeconomic and consumer characteristics to explore whether these characteristics would modify the relationship between price and demand for the food groups included in our analysis. We have stated this more clearly in Lines 109-113:

‘Price elasticities were first presented by pooling three survey rounds. We then calculated price elasticities stratified by: (1) survey round, (2) income quintile, (3) consumers of red meat (non-zero expenditures on red meat), and (4) consumers of processed meat (non-zero expenditures on processed meat) to estimate potentially heterogenous impacts on purchases associated with price increases on different consumer groups.’

2. While the conclusions are generally well-supported by the data, the practical feasibility of the substantial price increases proposed—especially to achieve dietary targets—could be discussed further to address potential consumer responses and implementation challenges.

Response: Thank you for this comment. Please see below more nuanced discussion on the feasibility, implementation, and potential consumer responses now included in Lines 387-413:

‘Our study taken together with the cumulative evidence suggest that large price increases are needed to achieve sufficient reductions in meat consumption, under a critical timeline if we want to meet urgent climate goals such as < 2�C degrees of warming (1) while addressing diet-related health burdens (38). As noted, imposing a tax only on processed meat may be more politically feasible given its stronger association with chronic disease, particularly colorectal cancer and diabetes (2,3,25,28), and that moderate consumption of red meat may still benefit lower income, indigenous populations as a source of nutrients, such as heme iron and vitamin B12 (8). To date, we are only aware of a beverage-specific tax that has been implemented at rates similar to those proposed for processed meat in this study, namely in Saudia Arabia and the United Arab Emirates (39). The food tax in Colombia is expected to increase to 20% by 2025 making it one of the highest food tax rates, globally (36). Factors such as public acceptance, feasibility, and implementation are important. Ensuring that dietary substitutes are available and affordable are critical for feasibility. Furthermore, several conditions may favor such a policy. Mexico currently has political momentum to pass progressive food legislation in favor of health and sustainability. Beyond incorporating sustainability into the national dietary guidelines as a policy tool to justify enforcement, Mexico implemented front-of-package labeling with nutrient warnings on packaged foods and beverages in 2020 (40), and has recently passed a ban on the sale of ultraprocessed foods and beverages in schools to be implemented in March 2025 (41). Additionally, Mexico enacted a General Law on Adequate and Sustainable Nutrition in 2024 that constitutionally protects the right to adequate, nutritious, and sustainably produced food (42), which may serve as a normative framework for complementary legislation in the future. The high suggested price increases in this study are warranted considering the concurrent environmental and health crises, as well as the Mexican government’s present political willingness to act on these issues. These findings serve as an ambitious scenario to understanding the tax rates needed to achieve sufficient changes in consumption across the Mexican population to meet health and environmental objectives. Ambitious targets may serve as a bargaining tool or escalate urgency, given that in practice, negotiation with government, industry, and the public often result in tax rates or policy measures that can fall short of what is deemed necessary by evidence (43–45).’

As well as in lines 424- 436:

‘On average, adherence to the 2023 Mexican Dietary Guidelines is less expensive than the current diet (by 21%) in the general population since costs are offset by red meat consumption. However, the 2023 Mexican Dietary Guidelines diet was more expensive than the current diet among lower income groups mostly due to an increase in expenditure on nuts, fruits, and vegetables (12). Therefore, special consideration should be taken to ensure the affordability of sustainable and healthy diets, including dietary substitutes, in this population. Again, taxing only processed meat does not pose a risk to diet quality, as its associated health risks outweigh potential benefits. Therefore, this approach may also help mitigate concerns regarding affordability of a ‘healthy’ diet among lower income populations. Nutrition surveillance of at-risk populations is critical to inform how compensatory measures may be designed to enhance dietary quality. Tax revenue recycling in the form of targeted lump sum transfers can be used to benefit lower income groups and counteract any potential regressivity of the tax (47), subject to Congress and government approval as, in Mexico, revenues are not earmarked.’

And lines 475- 489:

‘Investigation into social determinants of red and processed meat consumption could enhance the effectiveness of food tax policies. For instance, knowledge of within-country cultural differences could be leveraged to tailor targeted policies. Red meat consumption, for example, is higher in Northern and urban regions of the country (12) where it is more prominently featured in staple dishes and tied to culinary identity (52). However, there is reframing of this narrative in efforts seeking to revive the traditional Mexican diet – such as the 2023 Mexican Dietary Guidelines – as the popularity of traditional foods like beans has declined, while meat consumption has increased, particularly after the enactment of the North American Free Trade Agreement in 1994 (53,54). Expanding on these complementary public relations campaigns – such as those led by the Center for Research on Nutrition and Health in Mexico – could highlight food substitutes that are culturally tailored to regional dishes as well as the nutritional and environmental implications to encourage these dietary swaps. Moreso, research on whether lower income families will purchase and cook dietary substitutes is essential in addition to better understanding the nutritional implications of such reductions, ensuring that substitution patterns adequately deliver energy and nutrient requirements.’

3. The manuscript largely adheres to reporting standards, providing transparency in data sources and methodologies. However, while the statistical significance of the findings is frequently noted, there is insufficient discussion of their practical implications. Expanding on how these statistically significant results might translate into real-world outcomes would greatly enhance the reader's understanding of their relevance and potential impact.

Response: Thank you for raising this. This complements the response to your second point, which we have pasted a response for above.

We have also expanded on practical implications in discussing how this tax may have differential implications across socioeconomic groups [Lines 416-445].

‘Given that lower income groups allocate a smaller proportion of their food budget to meat – despite being more price sensitive – the tax would not necessarily impose a disproportionate burden on them. Moreover, lower income groups tend to have greater adherence to the sustainable and healthy traditional Mexican Diet, which serves as a foundation for Mexican National Dietary Guidelines (8,12). Therefore, red and processed meat consumption in this group would change by a smaller proportion, and food substitutions would align more closely with traditional dietary patterns as was observed in our models (12). In the case of health-motivated taxes, reductions in consumption among lower income groups can actually mitigate health disparities and healthcare expenditures (46). On average, adherence to the 2023 Mexican Dietary Guidelines is less expensive than the current diet (by 21%) in the general population since costs are offset by red meat consumption. However, the 2023 Mexican Dietary Guidelines diet was more expensive than the current diet among lower income groups mostly due to an increase in expenditure on nuts, fruits, and vegetables (12). Therefore, special consideration should be taken to ensure the affordability of sustainable and healthy diets, including dietary substitutes, in this population. Again, taxing only processed meat does not pose a risk to diet quality, as its associated health risks outweigh potential benefits. Therefore, this approach may also help mitigate concerns regarding affordability of a ‘healthy’ diet among lower income populations. Nutrition surveillance of at-risk populations is critical to inform how compensatory measures may be designed to enhance dietary quality. Tax revenue recycling in the form of targeted lump sum transfers can be used to benefit lower income groups and counteract any potential regressivity of the tax (47), subject to Congress and government approval as, in Mexico, revenues are not earmarked. They can be used to subsidize the cost of healthier, more nutrient-rich foods such as F&V and legumes especially since both food groups were stronger substitutes to red and processed meat in the lowest compared to highest income group. This was explored in a recent study evaluating the regressivity of meat taxes across Europe (47). It found that tax recycling through lump sum transfers to the lowest quintile were more equitable than lowering the price of fruits and vegetables (47). Moreover, a social-cost benefit analysis applied price elasticities obtained from a literature review to test price increases of 15 and 30% to total meat (including red, white and processed) and found that a 10% decrease in price of fruit and vegetables counteracted regressivity among lower income groups (17). In addition to financial incentives, nutritional education may be a complementary strategy to ensure that these populations are making the ‘healthier’ choice.’

And an additional caveat in Lines 345-346:

‘Therefore, increasing the price of red and processed meat may promote greater adherence to the guidelines, especially in lower income groups, which can improve diet quality.’

Reviewer #2

Response: We greatly appreciate the time the reviewer spent in evaluating the manuscript and providing the valuable suggestions below, which have contributed to the strengthening the impact of this analysis.

1. The manuscript needs further clarification and enhancement in some key areas. For example, the methodology, especially the estimation of price elasticities, would be enhanced by a detailed discussion of assumptions made during the analysis. This might include suggestions for investigating possible confounding variables-what affects consumer behavior other than price changes, such as cultural considerations or alternative sources for food products. This could also involve sensitivity analyses that prove the strength of the results found.

Thank you, in terms of technical assumptions made in the modeling, we have now provided more detail in Lines 99- 105:

‘The linear approximate model was chosen to model a linear relationship between price and quantity demanded for interpretability and consistency with prior food literature (20–22). The model was estimated using Ordinary Least Squares (OLS) equation and ‘other foods’ was omitted to account for potential collinearity between food groups. Estimation using OLS assumes that errors are normally distributed and that there is no correlation in errors between equations (20). We also restricted the model to comply with the homogeneity and symmetry constraints, and verified that the adding-up property was met (20).’

We have also conducted a sensitivity analysis estimating the linear approximate of the almost demand system with Seemingly Unrelated Regression instead of Ordinary Least Squares to validate the ro

---

## [Decision Letter · Decision Letter 1]

Dear Dr. Connors,

Thank you for submitting your manuscript to PLOS ONE. After careful consideration, we feel that it has merit but does not fully meet PLOS ONE’s publication criteria as it currently stands. Therefore, we invite you to submit a revised version of the manuscript that addresses the points raised during the review process.

**Please tackle all comments raised by reviewers.**

We look forward to receiving your revised manuscript.

Kind regards,

Charles Odilichukwu R. Okpala, PhD

Academic Editor

PLOS ONE

Additional Editor Comments:

Authors, thanks for your patience. Please, kindly examine the comments of reviewers, especially the one that recommended 'rejection'

It is important to make effort and elevate the quality of your work.

Reviewers' comments:

Reviewer's Responses to Questions

**Comments to the Author**

Reviewer #4: (No Response)

Reviewer #5: (No Response)

2. Is the manuscript technically sound, and do the data support the conclusions?

Reviewer #4: Yes

Reviewer #5: Yes

3. Has the statistical analysis been performed appropriately and rigorously?

Reviewer #4: N/A

Reviewer #5: Yes

4. Have the authors made all data underlying the findings in their manuscript fully available?

Reviewer #4: Yes

Reviewer #5: Yes

5. Is the manuscript presented in an intelligible fashion and written in standard English?

Reviewer #4: Yes

Reviewer #5: Yes

**Reviewer #4:**  Dear authhors

Manuscript title: Promoting adherence to the sustainable Mexican Dietary Guidelines through a tax on red and processed meat

My comment on the manuscript review was “This is an economic report, not a scientific paper” and this paragraph “We assessed how to achieve sustainable, equitable, and healthy dietary targets for red and processed meat consumption in Mexico. The four specific aims were to (1) estimate own- and cross-price elasticities of red and processed meat, (2) estimate demand responses to price increases in red and processed meat under various scenarios and apply them to consumption data, (3) explore if price elasticities varied by household income and other effect modifiers (i.e., survey year, and consumption status) and (4) corroborate cross-sectional findings by evaluating dietary substitutions and complements to red and processed meat in a longitudinal cohort of Mexican women” was copied from the manuscript to clear the objective of the present paper according to the opinion of the authors.

I appreciate these goals, but this study used only publicly available secondary data and did not include experimental work regarding Animal Physiology or meat technology.

This work presents academic information. Therefore, my opinion may not be correct according to my scientific trend. So, I am sorry to re-review this manuscript.

Regards

**Reviewer #5:**  This study makes a valuable contribution to the literature on food taxes and sustainable diets, particularly in the context of middle-income countries like Mexico. Its strengths lie in its robust methodology, focus on equity, and actionable policy recommendations. However, I would recommend minor revisions to strengthen the study before publication.

While the study highlights substitution patterns, it does not fully explore the potential nutrient gaps that might arise from reduced meat consumption, especially for vulnerable groups (e.g., iron and vitamin B12 deficiencies). A more detailed nutrient analysis would strengthen the study.

**Do you want your identity to be public for this peer review?** For information about this choice, including consent withdrawal, please see our Privacy Policy

Reviewer #4: No

Reviewer #5: **Yes: ** Peilin An

---

## [Author Response · Author response to Decision Letter 2]

14 Apr 2025

Manuscript Title: Promoting adherence to the sustainable Mexican Dietary Guidelines through a tax on red and processed meat

Thank you for providing further comments to the manuscript. We have kindly addressed them below. Please note that the line numbers refer to the manuscript version with untracked changes.

Reviewer #4

Dear authhors

Manuscript title: Promoting adherence to the sustainable Mexican Dietary Guidelines through a tax on red and processed meat

My comment on the manuscript review was “This is an economic report, not a scientific paper” and this paragraph “We assessed how to achieve sustainable, equitable, and healthy dietary targets for red and processed meat consumption in Mexico. The four specific aims were to (1) estimate own- and cross-price elasticities of red and processed meat, (2) estimate demand responses to price increases in red and processed meat under various scenarios and apply them to consumption data, (3) explore if price elasticities varied by household income and other effect modifiers (i.e., survey year, and consumption status) and (4) corroborate cross-sectional findings by evaluating dietary substitutions and complements to red and processed meat in a longitudinal cohort of Mexican women” was copied from the manuscript to clear the objective of the present paper according to the opinion of the authors.

I appreciate these goals, but this study used only publicly available secondary data and did not include experimental work regarding Animal Physiology or meat technology.

This work presents academic information. Therefore, my opinion may not be correct according to my scientific trend. So, I am sorry to re-review this manuscript.

We thank the reviewer for clarifying their original comments and are sorry that our scientific discipline’s approach differs from the expectations of the reviewer. This is an epidemiological study using nationally representative data and well- established, rigorous economics modeling. “Animal Physiology” and “meat technology” are not appropriate disciplines to answer the above-stated objectives on price elasticities, demand responses, and human consumption behavior.

Reviewer #5

This study makes a valuable contribution to the literature on food taxes and sustainable diets, particularly in the context of middle-income countries like Mexico. Its strengths lie in its robust methodology, focus on equity, and actionable policy recommendations. However, I would recommend minor revisions to strengthen the study before publication.

While the study highlights substitution patterns, it does not fully explore the potential nutrient gaps that might arise from reduced meat consumption, especially for vulnerable groups (e.g., iron and vitamin B12 deficiencies). A more detailed nutrient analysis would strengthen the study.

We greatly appreciate this input and agree that potential nutrient gaps are an important consideration. This was the motivation for our investigation into potential dietary substitutes and complements, given a diverse diet is an important predictor of micronutrient adequacy. For this reason, evaluating the entire diet is key. Recognizing this, the Mexican Dietary Guidelines emphasizes a diverse diet, replacing red and processed meat with beans, lentils, egg, poultry, and fish which we have evaluated here. Moreover, the guidelines have been designed to provide a micronutrient adequate diet. We have further elaborated on this in lines 345-348:

‘Therefore, increasing the price of red and processed meat may promote greater adherence to the guidelines, especially in lower income groups, which has been designed to deliver a micronutrient adequate diet, and therefore, may improve diet quality due to increases in consumption of nutrient-dense foods and reductions in discretionary foods.’

Additionally motivated by this point, we provided estimations regarding potential changes to consumption of food groups rich in iron and vitamin B12 – such as animal-sourced protein foods – to incorporate flexibility in real-life dietary substitutions. We found that consumption was predicted to increase following price increases to red and processed meat across all four scenarios due to substitution behaviors (Table 2).

However, given the heterogeneity of foods within these broad food groups and that we did not model changes in consumption of the rest of the diet, determining the specific changes in nutrient intakes is out of the scope of this research and has been emphasized as a direction for future research in lines 493-501:

‘Moreso, research on whether lower income families will purchase and cook dietary substitutes is essential in addition to better understanding the nutritional implications of such reductions, ensuring that substitution patterns adequately deliver energy and nutrient requirements. Moreover, further research into the nutrient implications resulting from reductions in red and processed meat – especially for iron and vitamin B12 – while considering substitutions in the rest of the diet is a key priority, especially among lower income populations. Adherence to the Mexican Dietary Guidelines and a diverse diet is indicative of micronutrient adequacy, not just red and processed meat dietary targets, therefore, this should be considered in future micronutrient evaluations.’

We have also added additional consideration on policies that be designed to promote the consumption of nutrient-rich foods (particularly those rich in vitamin B12 and iron) that target groups most at-risk of nutrient inadequacy such as food vouchers. Please see lines 438-441:

‘They can be used to subsidize the cost via subsidies, food vouchers, or cash transfers to enable purchasing of healthier, more nutrient-rich foods such as F&V and legumes, especially since both food groups were stronger substitutes to red and processed meat in the lowest compared to highest income group.’

And lines 447-452:

‘Food vouchers may facilitate more direct targeting of subgroups most at-risk of nutrient inadequacy and be designed for the purchase of nutrient-rich foods, particularly rich in iron and vitamin B12, such as seafood, poultry, or eggs. In a randomized study conducted in Ecuador, food vouchers were found to increase dietary diversity as opposed to in-kind food or cash assistance[48]. In addition to financial incentives, nutritional education may be a complementary strategy to ensure that these populations are making the ‘healthier’ choice.’

Our final point is that we also found that lower income groups would have less ‘drastic’ changes to their diet since they currently purchase less red and processed meat compared to higher income groups in Mexico as discussed in lines 418-424:

‘Given that lower income groups allocate a smaller proportion of their food budget to meat – despite being more price sensitive – the tax would not necessarily impose a disproportionate burden on them. Moreover, lower income groups tend to have greater adherence to the sustainable and healthy traditional Mexican Diet, which serves as a foundation for Mexican National Dietary Guidelines [8,12]. Therefore, red and processed meat consumption in this group would change by a smaller proportion, and food substitutions would align more closely with traditional dietary patterns as was observed in our models [12].’

---

## [Decision Letter · Decision Letter 2]

Dear Dr. Connors,

Thank you for submitting your manuscript to PLOS ONE. After careful consideration, we feel that it has merit but does not fully meet PLOS ONE’s publication criteria as it currently stands. Therefore, we invite you to submit a revised version of the manuscript that addresses the points raised during the review process.

**ACADEMIC EDITOR:**

We look forward to receiving your revised manuscript.

Kind regards,

Charles Odilichukwu R. Okpala, PhD

Academic Editor

PLOS ONE

Journal Requirements:

**Additional Editor Comments:**

The authors evaluated how price increases to red and processed meat could shift consumption for meat as well as other key food groups, and did this by using data from the Mexican National Household Income and Expenditure Survey (2018, 2020, 2022). The revisions done thus far has elevated the quality of the work. However, the editor suggests further revisions are needed before it can be accepted.

a) After several reads, I deeply worry that the title does not strongly reflect the objective of the work as stated above. I suggest title: Price increases of red and processed meat - Would it shift consumption to other foods?

Authors, think deeply about this, I believe the simpler the title, the better it is to your work. Also, this title will attract readers around the globe, plus citations.

b) The introduction needs to be totally revised. Many aspects of current version needs to be removed. Kindly follow this path, to build three strong paragraphs. Paragraph one should introduce price elasticity applied to food produce, what is the global debate, are they agreements, and disagreements, which food produce are more debated, make sure to mention those of animal food products. Paragraph two should then narrow down to price elasticity arguments as it relates to meat products, what are the sustainability implications, (you can mention emissions, healthy diet, government policy but it must be on a global perspective, and not narrowed to any country specifically. Paragraph 3 should then introduce Mexico, population, meat production, and existing policies that have helped accumulate government data. Then, create the hypothesis and gap in existing knowledge (paucity of published information regards how price increases to red and processed meat could shift consumption for meat as well as other key food groups. This is a more robust approach to state why estimates own- and cross-price elasticities could help provide better understanding of the accumulated government data.

c) Please, your materials and methods should start with a new subsection captioned "Schematic overview of evaluation program", which must have a flow diagram that shows the major steps of methods, and at least 4 sentences. Sentence one must introduce the flow diagram. Sentence two presents the major steps, and connects it to the overall aim of the work. Sentence three explains why this method is robust, and make sure you support it with relevant literature. Sentence 4 clarifies the steps taken to ensure validity and authenticity of the work.

d) I examined the remaining aspects of the methods. It is very ok. Make sure you end the methods with a new subsection "Statistical analysis" ...provide ample detail for this, show all the formulae for price estimate procedures, provide supporting references, please provide ample detail that is very sufficient, and convincing

e) I examined the results, it is very good. There should be more information from Price elasticities of meat by cut quality, even though you did not simulate demand responses to price increases by cut of meat.

f) Please, have another look at your discussion, and keep the discussion, strictly discussion. All the tables and figures captured in results, must be referred to in the discussion. Use (Refer to Table ?) or (Refer to Figure ?) where and when a specific data of table or figure is mentioned. I can see you have a lot of supplementary information. I suggest that you develop a figure that tactfully captures key message of various supplementary info, which should be used to support your discussion.

g) please, remove all aspects of conclusion, and create a section 'Concluding remarks", and move them there. please, remove all aspects of future studies, and move them to a new section 'Recommendations for future work"

Look forward to your revised manuscript. I believe the quality of your work will be far much better after this revision. Thank you and God bless

Reviewers' comments:

Reviewer's Responses to Questions

**Comments to the Author**

Reviewer #5: All comments have been addressed

2. Is the manuscript technically sound, and do the data support the conclusions?

Reviewer #5: Yes

3. Has the statistical analysis been performed appropriately and rigorously?

Reviewer #5: Yes

4. Have the authors made all data underlying the findings in their manuscript fully available?

Reviewer #5: Yes

5. Is the manuscript presented in an intelligible fashion and written in standard English?

Reviewer #5: Yes

Reviewer #5: (No Response)

**Do you want your identity to be public for this peer review?** For information about this choice, including consent withdrawal, please see our Privacy Policy

Reviewer #5: **Yes: ** Peilin An

---

## [Author Response · Author response to Decision Letter 3]

27 May 2025

Proposed Revised Title: Taxes to red and processed meat to promote sustainable and healthy diets in Mexico

We greatly appreciate the Editor’s thorough review of work and comments that have elevated the quality of the work. We have responded to comments line-by-line below. Line numbers refer to the version with untracked changes.

Additional Editor Comments:

The authors evaluated how price increases to red and processed meat could shift consumption for meat as well as other key food groups, and did this by using data from the Mexican National Household Income and Expenditure Survey (2018, 2020, 2022). The revisions done thus far has elevated the quality of the work. However, the editor suggests further revisions are needed before it can be accepted.

a) After several reads, I deeply worry that the title does not strongly reflect the objective of the work as stated above. I suggest title: Price increases of red and processed meat - Would it shift consumption to other foods?

Authors, think deeply about this, I believe the simpler the title, the better it is to your work. Also, this title will attract readers around the globe, plus citations.

Thank you for the title suggestion. The primary aim of our study was to explore how policies – specifically fiscal policies – could be used to achieve sustainable and healthy dietary targets in the Mexican population. While we were interested in shifts in consumption of other food groups to give us an understanding of potential changes to the entire diet, our primary focus was not whether these would shift (which was expected in the demand system) but how to promote adherence to specific dietary targets.

To better reflect this focus as well as expand readership, we propose the following revised title:

‘Taxes to red and processed meat to promote sustainable and healthy diets in Mexico’

b) The introduction needs to be totally revised. Many aspects of current version needs to be removed. Kindly follow this path, to build three strong paragraphs.

Paragraph one should introduce price elasticity applied to food produce, what is the global debate, are they agreements, and disagreements, which food produce are more debated, make sure to mention those of animal food products.

Paragraph two should then narrow down to price elasticity arguments as it relates to meat products, what are the sustainability implications, (you can mention emissions, healthy diet, government policy but it must be on a global perspective, and not narrowed to any country specifically.

Paragraph 3 should then introduce Mexico, population, meat production, and existing policies that have helped accumulate government data. Then, create the hypothesis and gap in existing knowledge (paucity of published information regards how price increases to red and processed meat could shift consumption for meat as well as other key food groups. This is a more robust approach to state why estimates own- and cross-price elasticities could help provide better understanding of the accumulated government data.

Thank you for the suggested revisions to the introduction. We have restructured the introduction as per request while maintaining emphasis on the objective of this study and food taxes rather than the methods employed to address the research question. Please find the revised introduction in Lines 53-94:

‘Government-supported demand-side interventions, such as fiscal policies, are one way to support food system transformations and behavior change [1]. Food-specific taxes have been most commonly applied to sugar-sweetened beverages (SSBs), nonessential energy-dense foods, and saturated fat to address diet-related chronic disease burdens [2–4]. Taxes have indeed been effective in reducing targeted food purchases [4,5], therefore, may serve as a policy tool to achieve double-duty goals for health and sustainability. Estimating price elasticities enhances empirical understanding of demand responses to price increases of taxed commodities as well as potential substitutes and complements to those commodities, and thus have been calculated to inform tax policy design [6,7].

Meat production accounts for a majority of food system emissions, accounting for one third of greenhouse gas emissions [8]. Simultaneously, high consumption of red and processed meat is also associated with colorectal cancer [9] and type 2 diabetes [10], among other health risks [11,12]. Moderating levels of meat consumption is central to sustainable and healthy diets [13,14], particularly in high- and middle- income contexts where meat consumption is disproportionately high [15,16]. Several studies have estimated or utilized price elasticities to model hypothetical taxes on meat in Europe, the United States, and more recently, Brazil for precisely this purpose [17–21]. Demand responses to price vary by population and commodity; curbing demand for red and processed meat has been found to require more aggressive approaches such as higher rates of taxation as well as varying rates across consumer groups and geographies [17,18]. Thus, derivation of price elasticities to estimate demand responses to such policies is a practical tool that may inform the magnitude of increases that would be needed to achieve desired consumption levels.

Global diets require drastic transformations to keep within net zero emissions targets and alleviate significant diet-related disease burdens [8,22]. Recognizing this, Mexico integrated sustainability into the national Dietary Guidelines [14]. Nevertheless, adherence to sustainable and healthy diets remains low in Mexico [23,24] despite research suggesting that such diets are affordable in this context [25,26]. In particular, meat consumption continues to increase and is well above recommended levels [23]. Thus, reducing red and processed meat consumption has been identified as key to increasing the sustainability of healthy diets in the Mexican context [23,24]. Mexico was one of the first countries to pass a food-specific tax in 2014 on SSBs and an ad valorem tax on non-essential energy-dense foods motivated by increasing diet-related chronic disease burdens such as obesity and type 2 diabetes [5]. The path toward a sustainable and healthy diet in Mexico has been clearly outlined [14] and assessed for feasibility and affordability [26], but an implementation gap remains that necessitates government policy intervention. Building on Mexico’s food policy toolkit [27], a tax to red and processed meat can help address this gap. To date, such a tax has not been studied in the Mexican context and understanding of how it might shift consumption in the rest of the diet -- essential to diet quality -- is not well understood.

We assessed how to achieve sustainable, equitable, and healthy dietary targets for red and processed meat consumption in Mexico. The four specific aims were to (1) estimate own- and cross-price elasticities of red and processed meat, (2) estimate demand responses to price increases in red and processed meat under various scenarios and apply them to consumption data, (3) explore if price elasticities varied by household income and other effect modifiers (i.e., survey year, and consumption status) and (4) corroborate cross-sectional findings by evaluating dietary substitutions and complements to red and processed meat in a longitudinal cohort of Mexican women.’

c) Please, your materials and methods should start with a new subsection captioned "Schematic overview of evaluation program", which must have a flow diagram that shows the major steps of methods, and at least 4 sentences. Sentence one must introduce the flow diagram. Sentence two presents the major steps, and connects it to the overall aim of the work. Sentence three explains why this method is robust, and make sure you support it with relevant literature. Sentence 4 clarifies the steps taken to ensure validity and authenticity of the work.

Thank you for this helpful suggestion. We would like to clarify that our study did not evaluate a program but rather modeled the potential impact of a food policy intervention. Thus, referring to a ‘schematic overview of evaluation program’ can be very misleading to readers. Please let us know if you have any comments in follow-up.

d) I examined the remaining aspects of the methods. It is very ok. Make sure you end the methods with a new subsection "Statistical analysis" ...provide ample detail for this, show all the formulae for price estimate procedures, provide supporting references, please provide ample detail that is very sufficient, and convincing

Thank you, we have now included a section titled, ‘Statistical Analyses’, with model specifications and price index estimation in Lines 118-169.

‘We calculated uncompensated own- and cross-price elasticities of demand for 10 food groups through estimation of a complete demand system using a linear approximate of an Almost Ideal Demand System (LA/AIDs)[28]. The linear approximate model was chosen to model a linear relationship between price and quantity demanded for interpretability and consistency with prior food literature [28–30]. Briefly, the model calculates a set of estimates based on the proportion of the budget dedicated to a particular food group out of the total household food expenditures. These estimates approximate the response in demand to changes in the price of a food group without considering changes in income.

Price was calculated based on the quantity purchased (kg or liters) and the expenditures reported for that quantity of the food group. Prices were then averaged at the municipality level to reduce potential record biases at the household level [6]. Outliers at the municipality level that fell ±2 standard deviations from the national mean were replaced with the weighted average national price for that food group (n=276 [0.4%] for 2018; n=166 [0.2%] for 2020; and n=138 [0.2%] for 2022). The expenditure share of each food group was calculated as the sum of expenditures in each food group divided by total household food expenditure.

The LA/AIDs model is defined as:

w_hgmt = α_g+ ∑_(j=1)^j▒〖β_gi ln⁡〖p_mjt 〗 〗 + γ ln E/(P*) + ∑_(v=1)^v▒〖δ_gv η_hmtv 〗 + μ_hmgt

LA/AIDs model Covariate term

As specified, w_ghmt is the proportion of household expenditures spent on a particular food group g for household h residing in municipality m in survey year t. Each individual food item within each food group is denoted as j and p_mjt represents the price of each food item at the municipality level m in the year t the survey was administered. E represents total household expenditures on food while lnP* is the Laspeyres price index represented as lnP_jt in Equation 1 above. The covariate term η denotes shifter variables which may affect demand for a good, v, at the household and municipality level that will be added to the model. Lastly, μ is the error term for the model.

We included the Laspeyres Price Index to account for changes in price over time and maintain linearity in the parameters by weighting price according to baseline prices and shares (in this case, 2018) for a constant basket of foods[31]. The index is calculated as:

lnP_jt = ∑_(g=1)^(j-1)▒(w_g ) ® * ln p_mjt

p_mjt is defined as the price per unit for each jth food item in municipality m at survey year t. In addition, (w_g ) ® is defined as the average budget share for the food group g pertaining to food item j. More detailed information on the price elasticity estimation can be found in S1 Appendix.

The model was estimated using Ordinary Least Squares (OLS) equation and ‘other foods’ was omitted to account for potential collinearity between food groups. Estimation using OLS assumes that errors are normally distributed and that there is no correlation in errors between equations [28]. We also restricted the model to comply with the homogeneity and symmetry constraints, and verified that the adding-up property was met [28]. We accounted for potential sociodemographic factors that could influence the relationship between price and demand through accounting for demand shifters in the model. These included educational attainment of the head of household, year of survey (2018, 2020, and 2022), region of residence (urban vs rural), and adult equivalent based on estimations derived from the Mexican population considering age and size composition of the household [33]. Adult equivalent has been previously included in food demand system analyses for Mexico [6] to account for how household composition may impact welfare due to household composition and the distribution of income amongst members. Price elasticities were first presented by pooling three survey rounds. We then calculated price elasticities stratified by: (1) survey round, (2) income quintile, (3) consumers of red meat (non-zero expenditures on red meat), and (4) consumers of processed meat (non-zero expenditures on processed meat) to estimate potentially heterogenous impacts on purchases associated with price increases on different consumer groups. More details on model specifications are in S1 Appendix. We used the package for Demand Analysis with the Almost Ideal Demand System in R statistical software to estimate the demand system [34].’

e) I examined the results, it is very good. There should be more information from Price elasticities of meat by cut quality, even though you did not simulate demand responses to price increases by cut of meat.

Thank you for this suggestion, we have now included more information on the cross-price elasticities with cuts of meat quality in Lines 272-286.

‘The own-price elasticities for cheaper and more expensive cuts of meat were -0.83 and -0.78, respectively (S2 Appendix; S16-S17 Tables; S2 Fig). Eggs and F&V were complements to cheaper cuts of meat whereas they were substitutes for expensive cuts of meat. Seafood and poultry were substitutes for cheaper cuts of meat but complements to expensive cuts. Grains and F&V were stronger substitutes to expensive cuts of meat compared to cheaper cuts and legumes were a stronger substitute to cheaper cuts. Stratified by income quintile, cheaper cuts were more price elastic than expensive cuts in the lowest income group whereas in the top income quintiles the opposite was true (S2 Fig.). Additionally, there were differential cross-price elasticities by income group (S17 Table). Poultry and dairy were stronger substitutes and eggs, discretionary foods, grains, and expensive cuts of meat were stronger complements to cheaper cuts of meat in the highest compared to lowest income group. Whereas, eggs and grains were stronger substitutes, and dairy, discretionary foods and cheaper cuts of meat were stronger complements to expensive cuts of meat in highest compared to lowest income group.

Cheaper cuts of meat tended to be processed meat whereas more expensive cuts of meat tended to be red meat (hence, the similarity between own-price elasticities) (S4 Table). Therefore, we did not simulate demand responses to price increases by cut of meat.’

f) Please, have another look at your discussion, and keep the discussion, strictly discussion. All the tables and figures captured in results, must be referred to in the discussion. Use (Refer to Table ?) or (Refer to Figure ?) where and when a specific data of table or figure is mentioned. I can see you have a lot of supplementary information. I suggest that you develop a figure that tactfully captures key message of various supplementary info, which should be used to support your discussion.

We are regular readers of PLOS One publications, and have reviewed several recently published articles, and have not seen tables, figures, and supplemental materials explicitly called out in this way in the discussion section. We are happy to do this based on your request, but we wanted to confirm that you wanted us to deviate from what appears to be standard practice to date for PLOS One papers first.

g) please, remove all aspects of conclusion, and create a section 'Concluding remarks", and move them there. please, remove all aspects of future studies, and move them to a new section 'Recommendations for future work"

We have now labeled these sections accordingly as suggested, thank you.

Look forwa

---

## [Editor Report · Decision Letter 3]

Taxes to red and processed meat to promote sustainable and healthy diets in Mexico

PONE-D-24-41175R3

Dear Dr. Connors,

We’re pleased to inform you that your manuscript has been judged scientifically suitable for publication and will be formally accepted for publication once it meets all outstanding technical requirements.

Kind regards,

Charles Odilichukwu R. Okpala, PhD

Academic Editor

PLOS ONE

Additional Editor Comments (optional):

Accepted for publication
---

## [Editor Report · Acceptance letter]

PONE-D-24-41175R3

PLOS ONE

Dear Dr. Connors,

I'm pleased to inform you that your manuscript has been deemed suitable for publication in PLOS ONE. Congratulations! Your manuscript is now being handed over to our production team.

Kind regards,

on behalf of

Dr. Charles Odilichukwu R. Okpala

Academic Editor

PLOS ONE